# Brief communication: Visualizing uncertainties in landslide susceptibility modeling using bivariate mapping

Matthias Schlögl[1,2], Anita Graser[3], Raphael Spiekermann[4], Jasmin Lampert[3], and Stefan Steger[4]

[1]Department of Landscape, Spatial and Infrastructure Sciences, University of Natural Resources and Life Sciences, Peter-Jordan Straße 82, 1190 Vienna, Austria
[2]Department for Climate Impact Research, GeoSphere Austria, Hohe Warte 38, 1190 Vienna, Austria
[3]Center for Digital Safety & Security, Austrian Institute of Technology, Giefinggasse 4, 1210 Vienna, Austria
[4]RiskLab, GeoSphere Austria, Hohe Warte 38, 1190 Vienna, Austria

**Correspondence:** Matthias Schlögl (matthias.schloegl@boku.ac.at)

**Abstract.** Effectively communicating uncertainties inherent to statistical models is a challenging yet crucial aspect of the modeling process. This is particularly important in applied research, where output is used and interpreted by scientists and decision makers alike. In disaster risk reduction, susceptibility maps for natural hazards are vital for spatial planning and risk assessment. We present a novel type of landslide susceptibility map that jointly visualizes the estimated susceptibility and the corresponding prediction uncertainty, using an example from a mountainous region in Carinthia, Austria. We also provide implementation guidelines to create such maps using popular free and open-source software packages.

## 1 Introduction

In the context of disaster risk reduction, landslide risk assessment plays a pivotal role in identifying, assessing, understanding, managing, and mitigating the potential impacts of landslides on human lives, infrastructure, and the environment (Schlögl et al., 2019; Dai et al., 2002). Landslide susceptibility modeling comprises a set of computational approaches tailored towards the identification of areas exhibiting an increased likelihood of landslide occurrence (Guzzetti et al., 2006). In contrast to deterministic physical slope stability modeling, empirical statistical landslide susceptibility modeling usually employs methods of statistical learning applied in the context of a binary classification problem (Spiekermann et al., 2023). Statistically-based landslide susceptibility models use documented landslides from inventories as target variable (training labels) and a number of predisposing factors such as terrain conditions, lithology or land cover as independent variables. Having completed the model tuning, training and prediction process, the estimated class probability of the positive class is referred to as landslide susceptibility (Reichenbach et al., 2018). Thus, areas with high probabilities exhibit similar characteristics to landslide locations in the past and are therefore assumed to be more susceptible to slope instability during future triggering events.

Despite the generally acknowledged utility of landslide susceptibility models in certain contexts, their impact is contingent on the input data quality (Loche et al., 2022; Lima et al., 2021; Gaidzik and Ramírez-Herrera, 2021; Steger et al., 2016a) and geomorphic plausibility of the results (Steger et al., 2021). Geomorphic plausibility evaluation aims to assess whether a landslide susceptibility map aligns with fundamental process knowledge or rather reflects errors stemming from input data or

the modeling approach, as detailed in Steger et al. (2016b). On the other hand, overall usability of maps and interpretability of the underlying models is another aspect influencing the practical applicability. Thus, communicating complex scientific findings effectively is crucial not only for scientific advancement but also for decision-making in policy and practice (Weigold, 2001).

In this context, a particularly crucial aspect of science communication is the uncertainty associated with scientific methods and findings (Schneider, 2016). Yet uncertainty evaluation is often underrated despite having potentially far-reaching implications for decision makers needing to interpret the information on landslide susceptibility. Uncertainty is inherent to science as it describes the quality and reliability of observations and models and, consequently, also relevant to science communication (Gustafson and Rice, 2020). Effectively conveying uncertainty is essential for upholding trust in research and supporting successful user uptake of findings (Fischhoff and Davis, 2014). This is particularly the case in the context of numerical modeling of natural systems, where verification and validation is impossible and the primary value of models is heuristic (Oreskes et al., 1994).

In the area of landslide susceptibility modeling, different sources of uncertainty have been investigated. This includes, among others, the error propagation of inventory-based positional errors (Steger et al., 2016a), the effects of different landslide boundaries and spatial shape expressions (Huang et al., 2022), the effect of spatial resolution (Huang et al., 2023), the number of non–landslides sampled (Hong et al., 2019) and different resampling strategies (Moreno et al., 2023).

While quantification of model fit and model performance has become standard practice (Reichenbach et al., 2018), the vast majority of published landslide susceptibility maps do not spatially compute the uncertainty associated with the estimated susceptibility. As the uncertainty of landslide susceptibility predictions can be highly variable in space, it is imperative that decision makers have access to spatial information describing the uncertainty. A small number of studies used ensemble modeling as means to quantify uncertainty, employing metrics for statistical dispersion on the ensemble predictions to quantify uncertainty. Rossi et al. (2010) suggested that the combination of landslide susceptibility zonations based on multiple forecasts can improve the quality of susceptibility assessments. In their comparative model evaluation study they provided maps of the geographical distribution of the model error across slope units. Petschko et al. (2014) provided a comprehensive analysis of model uncertainty based on the standard error of model prediction in their quality assessment of landslide susceptibility maps, and also presented a spatial visualization of map uncertainty. Lombardo et al. (2014) accounted for the dispersion of the susceptibility estimates obtained by training a series of models using different partitioning strategies and reported the results in a model error map, using the two standard deviations interval. Heckmann et al. (2014) used the interquantile range (between the 0.95-quantile and the 0.05-quantile) of 100 susceptibility maps derived via random resampling as a non-parametric measure of dispersion to quantify the uncertainty caused by sampling and stepwise model selection in their debris flow susceptibility modeling assessment. Peng et al. (2014) used rough set support vector machine model trained on five different random samples to obtain estimates of landslide susceptibility for all mapping units. They also used two standard deviations as metric to quantify model uncertainty. Achu et al. (2023) employed an ensemble of different machine learning models and visualized the uncertainty stemming from the model types using the coefficient of variation across the predicted probabilities of all ensemble members.

Against the backdrop of the diverse methodologies available to account for uncertainties, scientists face challenges in effectively communicating their results, while decision makers struggle to interpret this complex information. Although the challenge of representing and managing geospatial information uncertainty is well understood in geoinformation science, particularly in cartography, this awareness is not as pervasive in other science disciplines and application areas, including geomorphology. Various methods for visualizing geospatial information uncertainty have been presented and discussed (Kinkeldey et al., 2014; MacEachren et al., 2005), including applications in slope stability modeling (Davis and Keller, 1997). All previously mentioned studies on landslide susceptibility modeling reported results by including separate maps for the estimated susceptibility and uncertainty. In this study we visualize the uncertainty caused by the sampling of negative instances by showing the variability of the predictions conditional on sampling in space, utilizing an ensemble of random forest models. We present a novel type of visualizing landslide susceptibility maps that jointly represents the susceptibility estimate along with the corresponding uncertainty. We advocate that bivariate mapping is a straightforward yet sound and effective way to communicate landslide susceptibility and the associated uncertainty. We provide implementation guidelines to create such maps using free and open-source software (FOSS), with examples in R and QGIS.

## 2 Methods

### 2.1 Landslide susceptibility modeling

The study region encompasses an area of $5,785$ km$^2$ within the Central Eastern Alps in Carinthia, Austria. A total number of $1,973$ shallow landslides were documented in the landslide inventory compiled for the study region. These events served as target variable (i.e., training labels). A wide range of predisposing factors were considered as independent variables, including indicators describing geomorphology, hydrology, climatology, lithology, land cover, surface runoff and transportation infrastructure. A subset of 21 features was eventually distilled from the total set of 46 features initially identified as potentially relevant determinants for landslide occurrence after an iterative feature selection process. Feature selection was conducted based on two main considerations. First, highly correlated features were dropped to ease interpretability of feature effects. Second, features effectively representing potential inventory biases were removed.

We used a pixel-based approach at a spatial resolution of 10 meters. Given the large study area and the comparatively small number of landslides, this resulted in an event rate of only $2.7 \cdot 10^{-5}$. Therefore, we used random downsampling in the context of balanced bagging to counter the severe class imbalance present in the data set. Negative instances were randomly sampled using probability-proportional-to-size sampling. The absence sampling area was constrained to avoid sampling in trivial (e.g., lakes) and problematic (e.g., too close to existing slides, high-elevation regions such as glaciers exhibiting different process characteristics) areas. In order to obtain a balanced data set, the number of negative instances sampled was identical with the number of positive samples present in the data set. In order to quantify sampling uncertainty, we repeated the sampling process 10 times. Since all positive instances were kept in all iterations, the sampling variation stems entirely from the negative instances. There are more robust methods available that also account for variations in landslide presence data (Lima et al., 2023; Pourghasemi et al., 2020). The resulting 10 data sets served as a basis to train random forest models using nested

spatial resampling for model tuning and performance evaluation. Hyperparameter tuning was conducted using model based optimization.

This procedure eventually resulted in an ensemble consisting of 10 models trained and evaluated by means of nested spatial cross validation. Results of the single models were aggregated using the ensemble mean as an estimate of predicted susceptibility, and the ensemble standard deviation as to quantify the corresponding sampling uncertainty of the model. A more detailed description of the modeling approach as well as an in-depth discussion focusing on statistical performance and geomorphic plausibility is provided in Schlögl et al. (2025).

## 2.2 Visualization

### 2.2.1 Concept

The theory and applications of bivariate (choropleth) maps are well established (De Cola, 2002; Nelson, 1999; Trumbo, 1981), including applications in climatology (Teuling et al., 2011) and hydrology (Speich et al., 2015). A seminal and easily accessible introduction to bivariate choropleth maps was provided by Joshua Stevens in 2015 (Stevens, 2015). We build upon these considerations and apply the general principle to landslide susceptibility modeling, using the ensemble mean and the ensemble standard deviation of predicted probabilities as variables for the bivariate mapping of landslide susceptibility and corresponding uncertainty.

We utilize a classical $3 \times 3$ bivariate visualization, resulting in nine classes in total. The reclassification method employed to stratify the continuous outputs into classes depends on the feature:

- **Susceptibility (ensemble mean)**: Landslide susceptibility maps are commonly created using probabilistic binary classifiers, with the predicted outcomes naturally occurring on a continuous scale within the interval $[0, 1]$. While discretizing continuous variables poses certain challenges, such as the loss of information and certain pitfalls caused by the artificial breakpoints, categorizing continuous outcomes can aid in conveying results to stakeholders. Inspired by the methodology of Spiekermann et al. (2023), we established two thresholds based on the descending rank order plot, utilizing the empirical 0.8 and 0.95 quantiles of the predicted landslide susceptibility values for positive instances to separate the three susceptibility categories. This effectively means that 80 % of the observed events fall into the highest class, 15 % of the events are contained in the medium class, and only 5 % of the events fall in the lowest class. This reclassification method was chosen due to the straightforward and intuitive interpretability. The thresholds resulting from the two selected quantiles are 0.4481 and 0.6096, respectively.

- **Uncertainty (ensemble standard deviation)**: Given the skewness of the standard deviation, we used quantile-based reclassification to split the continuous variable into three classes. In the present case, this corresponds to the thresholds 0.0297 and 0.0416.

The bivariate susceptibility maps are based on two sequential single-hue color schemes, each increasing in saturation and value, with one scheme representing each variable. By blending these color schemes, a bivariate color palette emerges (Fig. 1).

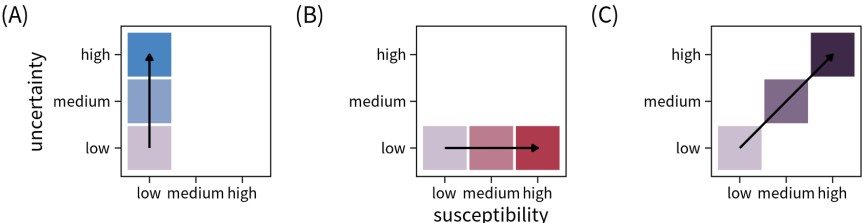

**Figure 1.** Conceptual illustration of a bivariate legend in the context of landslide susceptibility mapping. (A) shows the univariate color scheme for the uncertainty mapped to the y-axis, with uncertainty increasing from bottom to top. (B) displays the univariate color scheme for susceptibility along the x-axis, with susceptibility increasing from left to right. In both cases, single-hue color palettes are used, and higher values are characterized by higher saturation and value. (C) depicts the combined bivariate result after blending the two individual color palettes, with the diagonal showing joint increases in both variables.

We conducted stakeholder workshops using the World Café method (Löhr et al., 2020) with selected expert users from our main target group. Seven representatives from national civil protection and disaster management participated, including geologists from the Austrian National Geological Survey, geologists from Austrian federal governments, and representatives from disaster relief forces. They provided unstructured feedback on the use of the map in specific contexts and for specific applications, such as for spatial planning tasks. The World Café sessions were held on 27 February 2024 in Vienna under the aegis of the Disaster Competence Network Austria. Informed consent procedures were followed to ensure that stakeholders participated willingly and were adequately informed about the research. All statements were collected and summarized under the preservation of anonymity.

### 2.2.2 Implementation

There are several tools that support the practical implementation of the concept of bivariate susceptibility mapping. Here we exemplarily illustrate the implementation using two open source tools, representing one programming approach and one desktop GIS approach.

Bivariate susceptibility maps can be created programmatically in R using commonly used packages for managing geodata as well as the ggplot2 plotting framework. The package biscale provides convenience functions for reclassifying data into the desired number of classes and creating bivariate legends. It also offers a predefined set of bivariate color palettes. Technical details are provided in Appendix A, and all code is available in the supplementary GitHub repository.

Bivariate susceptibility maps can also be created using a classical desktop GIS. In QGIS, there are two main kinds of approaches to generating bivariate maps from rasters: (1) Combining two raster layers with blending modes (such as "multiply") to generate the bivariate effect or (2) reclassifying the raster into the desired number of classes using raster algebra. The Bi-

variate Polygon Renderer plugin can be used to create the legend in the print layout of the map. Technical details are provided in Appendix B, and a QGIS-project is available in the supplementary GitHub repository.

## 3  Results

The bivariate visualization of landslide susceptibility and associated uncertainty is exemplified for a small area for demonstrative purposes (Fig. 2). The map shows an area of the Nockberge mountain range of Carinthia. Lakes (most notably 'Brennsee' in the West) and high elevation areas ($> 1,900$ m) around the 'Wöllaner Nock' mountain (2,145 m a.s.l., East) are masked out since they are out of scope of the model (Schlögl et al., 2025).

The legend of the bivariate map is grid-like, with the four corner cases signifying the most extreme combination of the two variables 'susceptibility' and 'uncertainty'. The description and interpretation of the combined information holds irrespective of the number of classes used.

– **Lower left corner: low susceptibility and low uncertainty**. These areas exhibit low susceptibility with a high degree of confidence, with the model ensemble consistently indicating a low occurrence probability of shallow landslides. Thus, the occurrence of shallow landslides in these areas is unlikely. Yet, the predictions still need to be interpreted with care, given the limitations of the underlying data, including the landslide inventory used. Results should still be assessed in light of geomorphic plausibility of the results (Steger et al., 2016b), and the data basis as well as the modeling approach should be integrated in the assessment.

– **Lower right corner: high susceptibility and low uncertainty**. These areas are identified as highly susceptible to landslides with a high degree of confidence. This means that the data and model ensemble used to predict landslide susceptibility yield consistent and reliable results in these areas. Areas falling into this class could be candidates for prioritizing mitigation efforts, monitoring, and preventive measures.

– **Upper left corner: low susceptibility and high uncertainty**. This class represents areas that are identified as having low susceptibility to landslides, but the model prediction is associated with high uncertainty. In the case of an ensemble modeling approach this means that the ensemble members disagree, yielding a considerable spread of predicted probabilities across the single models. This suggests that the model's prediction of low susceptibility is less reliable, entailing that the high uncertainty warrants caution. While these regions could be considered to be relatively safe from landslides, which may in turn imply that resources for adaptation and mitigation measures might be better allocated elsewhere, these regions must not be simply neglected. Additional studies and data collection may be necessary to confirm the low susceptibility, especially if the area is important from a geomorphic point of view.

– **Upper right corner: high susceptibility and high uncertainty**. These areas are also identified as highly susceptible to landslides, but with a high degree of uncertainty. This indicates that, while the model suggests high susceptibility, the prediction is less reliable. Decision-makers should exercise caution and consider additional analyses or validation

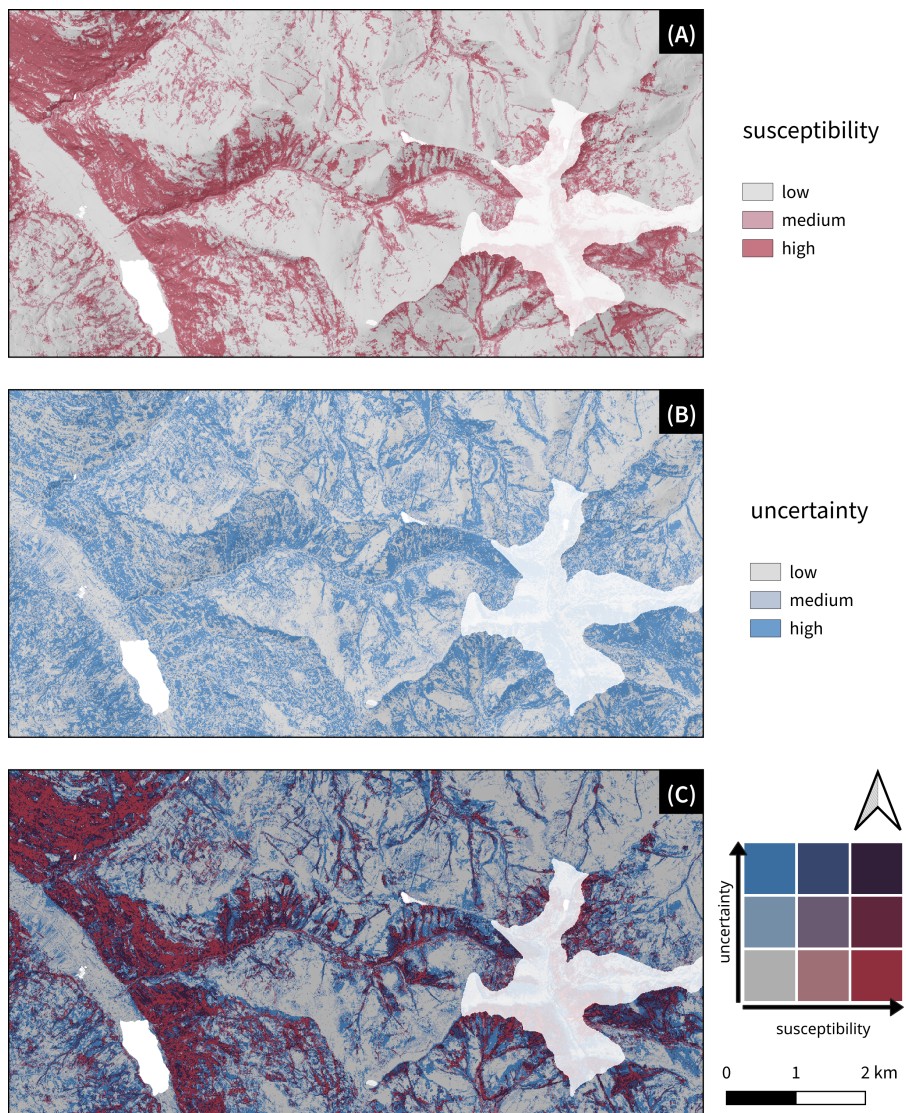

**Figure 2.** Example of a bivariate susceptibility map and its univariate components. (A) shows the susceptibility, (B) uncertainty and (C) the bivariate result using multiply blending. Lakes (transparency: 0 %) and high elevation areas (transparency: 20 %) are masked out.

before taking action. Areas within this category could be designated as priority areas requiring further investigation to reduce uncertainty. This can be accomplished by employing local model interpretation and diagnostic methods to explain these individual predictions (e.g., local surrogate models or individual conditional expectation curves), by investing in additional data collection or conducting field-based validation.

Counts within the resulting nine classes are not evenly distributed (Fig. 3). In the case of Carinthia, the class where both susceptibility and uncertainty is low constitutes the largest class, containing approximately 30 % of all pixels. Overall, areas with low susceptibility account for about 50 % of the study area.

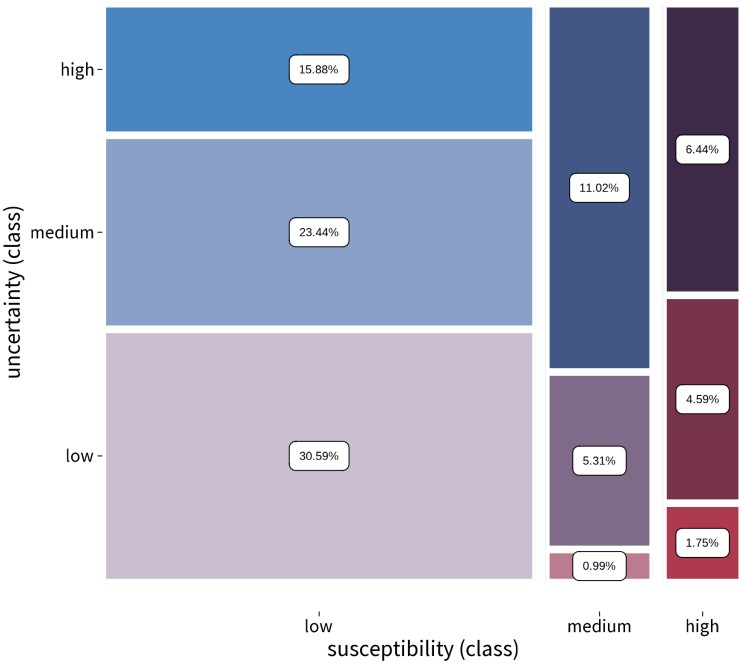

**Figure 3.** Mosaic plot displaying the frequency distribution of the nine bivariate classes of the full area of interest ($n = 46,760,232$). The plot is based on the contingency table of the classes and shows both the conditional and the marginal distribution of the class counts. The area is proportional to the cell frequencies of the underlying contingency table, and color is also mapped to the cell frequency.

Feedback on the usability of the bivariate map, as collected during the stakeholder workshops, indicated that users initially needed some time to become accustomed to the visualization. However, once the legend was internalized, the combined map was considered an effective means of communicating the integrated information. Two main main benefits over the use of two separate maps were identified. First, there is no need to switch between map views, making the joint interpretation less cumbersome. In the medium and long run, the cognitive effort required for internalizing the legend once is lower than that
required for switching between maps. Second, the consideration of the second dimension (uncertainty) was deemed less likely to be neglected or overlooked.

## 4 Discussion

There are two main properties that shape the appearance of the resulting map: (1) the definition of classes and (2) the color palette. In addition, the type of uncertainty conveyed should be kept in mind: The underlying uncertainties can be aleatoric

and epistemic in nature. Especially epistemic uncertainty, stemming from different sources along the modelling workflow, including the inventory, explanatory features or the model used, can be visualized.

Generally speaking, a wide range of methods are available for deriving univariate class intervals in continuous numerical vectors, including the use of equal intervals, quantile breaks, breaks derived through hierarchical or partitioning clustering methods, natural breaks optimization and algorithms for heavy-tailed distributions (Slocum et al., 2022; Jiang, 2013; Jenks and Caspall, 1971; Fisher, 1958). In landslide susceptibility modeling, results are commonly discretized into three classes signifying low, medium and high susceptibility, respectively. In addition to the aforementioned methods for deriving class intervals (Conoscenti et al., 2016; Hussin et al., 2016; Costanzo et al., 2012), modified methods for calculating breaks between susceptibility have been proposed, including methods based on the rank order (Spiekermann et al., 2023; Petschko et al., 2014; Chung and Fabbri, 2003) or on the receiver operating characteristic curve (Steger et al., 2024). The number of classes and the discretization algorithm used should be carefully considered and documented clearly, as these choices may have considerable impact on the appearance of the resulting map. We recommend limiting the number of classes to three or four in order to retain readability of the resulting bivariate classes in the map. The fact that the two variables are very likely to exhibit skewed distributions should also be taken into account when creating class intervals. Using easily understandable class definitions such as that proposed by Spiekermann et al. (2023) also aids interpretability: 80 % of all events occurred in the highest class, the medium class contained the next 15 % of observed events, and only the final 5 % of observed events fall into the lowest class.

The selection of a color palette is vital for map design, as it profoundly influences a map's effectiveness, readability and aesthetic appeal. Thus, choosing an appropriate color palette is essential to prevent errors or biases in data interpretation (Schloss et al., 2019; Seipel and Lim, 2017). This encompasses considerations of accessibility, especially for people with vision deficiencies (Nowosad, 2020)[1]. Additionally, the cultural and contextual relevance should be accounted for, recognizing the varying meaning of colors across cultures (Kawai et al., 2022) and the psychological implications as per the Color-in-Context theory (Elliot and Maier, 2012). Tailoring visualised information to users through a user-centered design process improves its effectiveness (Twomlow et al., 2022).

Landslide susceptibility models are associated with a wide range of epistemic (i.e., lack of knowledge) and aleatory (i.e., intrinsic randomness) uncertainties that propagate through the modeling chain (Knevels et al., 2023; Lombardo et al., 2020; Steger et al., 2016a; Brenning et al., 2015; Petschko et al., 2014; Rossi et al., 2010). The uncertainty displayed here merely refers to the estimation uncertainty stemming from the sampling of negative instances, as quantified through the ensemble modeling approach. However, the proposed bivariate depiction of landslide susceptibility is not contingent on specific types of uncertainties, and we advocate for a comprehensive inclusion of quantifiable uncertainties in such maps.

---

[1]Tools such as https://colorbrewer2.org or https://hclwizard.org, along with the corresponding R package `colorspace` (Zeileis et al., 2020), provide valuable assistance in selecting color palettes for users with different types of vision deficiencies.

## 5 Conclusions

We presented a bivariate landslide susceptibility map that jointly visualizes the estimated susceptibility and the corresponding uncertainty. This type of visualization is generally applicable to all different kinds of uncertainty in the context of susceptibility modeling for various natural hazards. The methodology can be easily implemented in popular FOSS packages using the examples provided in the supplementary repository as additional guidance. Overall, we argue that this graphical representation of susceptibility maps can aid in communicating modeling results and associated uncertainty more effectively.

By understanding the combinations of landslide susceptibility and associated uncertainty, informed decisions about where to focus efforts for data collection, monitoring, risk mitigation, and resource allocation can be supported. This approach ensures a balanced and strategic response to landslide risk management, addressing both the immediate and long-term needs based on the confidence level of the modeled susceptibility.

The research leading to this publication was partially carried out within the gAia project. The gAia project is funded by
230 the KIRAS program of the Austrian Research Promotion Agency (FFG) and the Federal Ministry of Agriculture, Regions and Tourism (BMLRT) under grant no. FO999886369.

*Code and data availability.* All code to generate the plots and maps is available on GitHub at https://github.com/r3xth0r/bivariate-lsm. Sample data is also provided in this repository via Git LFS. Code for the underlying landslide susceptibility model for Carinthia is available on GitLab at https://gitlab.com/Rexthor/lsm-carinthia.

*Author contributions.* M.S.: Conceptualization, Methodology, Software, Validation, Formal analysis, Data Curation, Writing - Original Draft, Writing - Review & Editing, Visualization; A.G.: Conceptualization, Methodology, Software, Visualization; S.S.: Validation, Writing - Review & Editing; J.L.: Conceptualization, Visualization, Writing - Review & Editing; R.S.: Validation, Writing - Review & Editing

*Competing interests.* The authors declare that they have no conflict of interest.

*Acknowledgements.* We thank Christina Rechberger and Susanna Wernhart from the Disaster Competence Network Austria (DCNA) for
their efforts in preparing and moderating the stakeholder workshops, as well as for compiling the acceptance analysis and exploitation concept.

## Appendix A: Bivariate susceptibility maps in R

The open-source programming language R (R Core Team, 2024) is a software environment for statistical computing and data visualization. It does also feature excellent support for geospatial data processing, geocomputational analyses and geographic information science (Wimberly, 2023; Lovelace et al., 2019; Pebesma, 2018). Visualization of bivariate maps in R is straightforward (Fig. A1) and requires the following add-on packages:

- `sf` (Pebesma et al., 2024a) is used to handle spatial vector data by extending data frames with a geometry column, thereby providing simple feature access (Pebesma and Bivand, 2023; Pebesma, 2018).

- `stars` (Pebesma et al., 2024b) is used to handle spatial raster data (Pebesma and Bivand, 2023).

- `ggplot2` (Wickham et al., 2024) provides a system for declaratively creating graphics based on 'The Grammar of Graphics' (Wickham, 2016; Wilkinson, 2005) and is used as a plotting framework.

- `biscale` (Prener et al., 2022) extends ggplot2 by providing tools and color palettes for bivariate thematic mapping.

- `ggspatial` (Dunnington et al., 2023) extends ggplot2 by providing additional support for plotting spatial objects.

- `patchwork` (Pedersen, 2024) or `cowplot` (Wilke, 2024) can be used to arrange multiple ggplot2 objects and compose a unified single plot.

- `rayshader` (Morgan-Wall, 2024) can be used for 3D visualizations of bivariate susceptibility maps.

Note that plotting raster data as points using `geom_raster()` is recommended for reasons of computational performance.

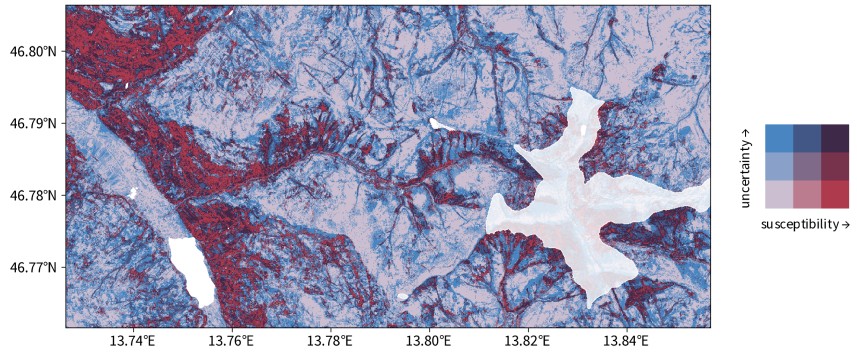

**Figure A1.** Example of a bivariate susceptibility map created in R with ggplot2. Legend created with the package biscale using the "DkViolet" color ramp created by Grossenbacher and Zehr (2019).

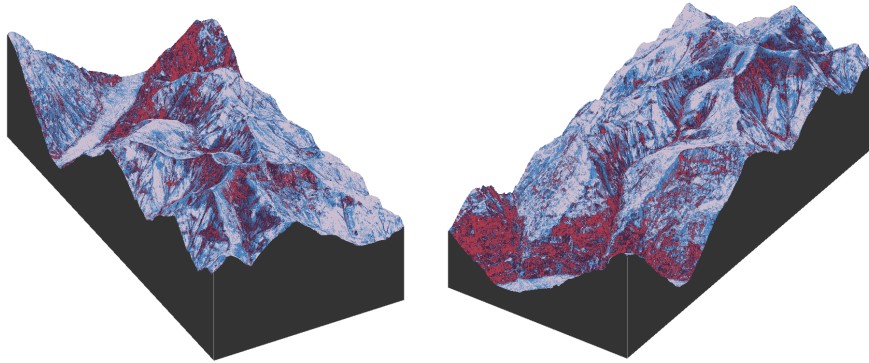

**Figure A2.** Example of a 3D visualization of the bivariate susceptibility map using rayshader. The figure displays to rendered shapshots. An interactive version is provided as Supplement 1.

## Appendix B: Bivariate susceptibility maps in QGIS

Existing QGIS plugins for bivariate mapping ('Bivariate Legend'[2] and 'Bivariate Polygon Renderer'[3]) focus on vector data and, therefore, do not support the creation of bivariate maps from susceptibility rasters. Albeit simply polygonizing the rasters and plotting the resulting polygons is possible, this does not scale well to large files.

There are two main kinds of approaches to generating bivariate maps from rasters: (1) Combining two raster layers with blending modes (such as "multiply") to generate the bivariate effect or (2) reclassifying the raster into the desired number of classes using raster algebra. From QGIS 3.22, the raster calculator supports IF statements[4], which make it straightforward to reclassify the the susceptibility raster into the classes required for the bivariate map.

The Bivariate Polygon Renderer is based on the blending approach with support for multiply, darken and mixing blending. It is worth noting however, that blending colors, as shown in Figure B1, does not result in the exact same bivariate color maps that can be seen in Figure A1 since the pink and blue colors are always blended with the grey, resulting in slightly darker tones.

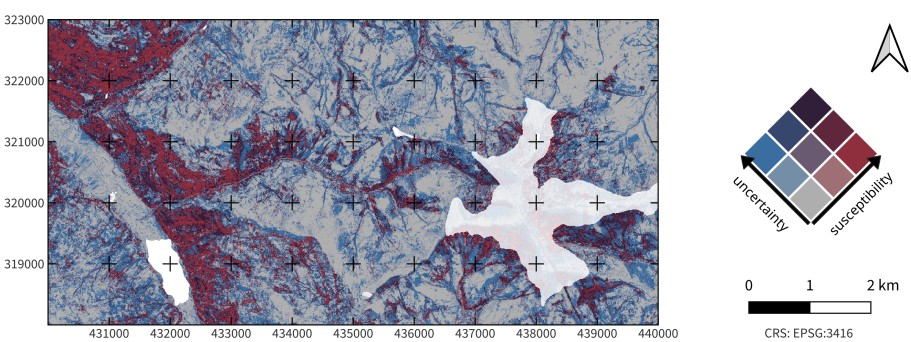

**Figure B1.** Example of a bivariate susceptibility map created in QGIS using multiply blending. Legend created using the Bivariate Renderer plugin with the "Violet - Blue" color ramp.

---

[2]https://plugins.qgis.org/plugins/BivariateLegend/

[3]https://plugins.qgis.org/plugins/BivariateRenderer/

[4]https://www.qgis.org/en/site/forusers/visualchangelog322/indeux.html#feature-add-if-function-to-raster-calculator

# Appendix C: Bivariate color palettes

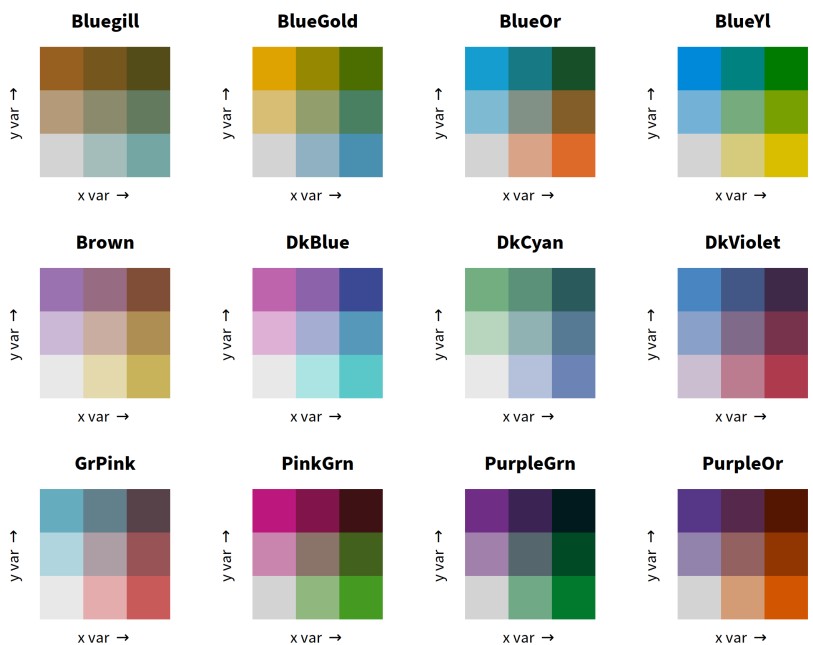

**Figure C1.** Overview of color palettes for bivariate maps as available in the R package `biscale`.

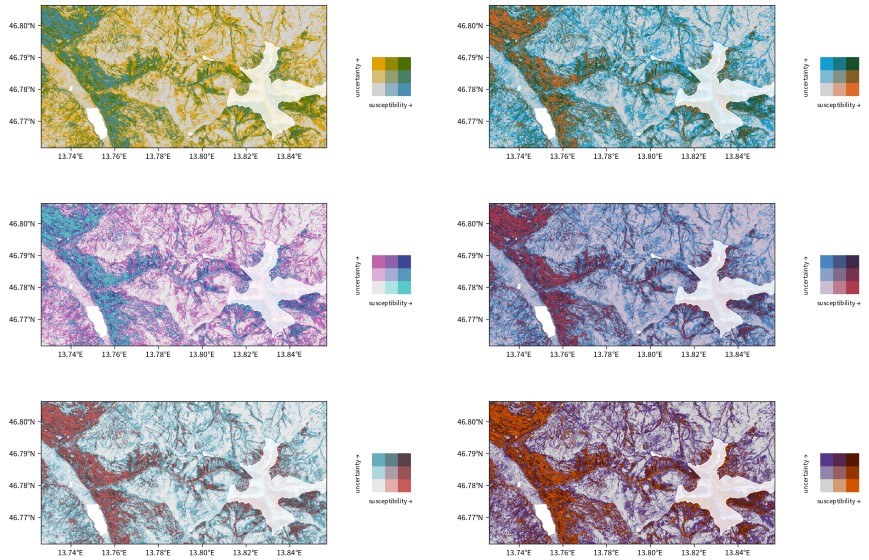

**Figure C2.** Illustrations of the bivariate susceptibility map using the color palettes "BlueGold", "BlueOr", "DkBlue", "DkViolet", "GrPink" and "PurpleOr".

## Appendix D: Conditional dependence of standard deviation on the mean

When interpreting the uncertainty estimate, the conditional dependence of the ensemble standard deviation on the mean susceptibility needs to be considered (Fig. D1). It can be observed that the uncertainty of the outcome has a maximum around 0.5 to 0.6 and decreases towards both ends of the susceptibility spectrum, exhibiting an approximately quadratic relationship which can be approximated as $y = 0.1745x - 0.1658x^2$. Given the high heteroscedasticity, this relationship was estimated using robust regression with M estimation, employing Tukey's bisquare loss function.

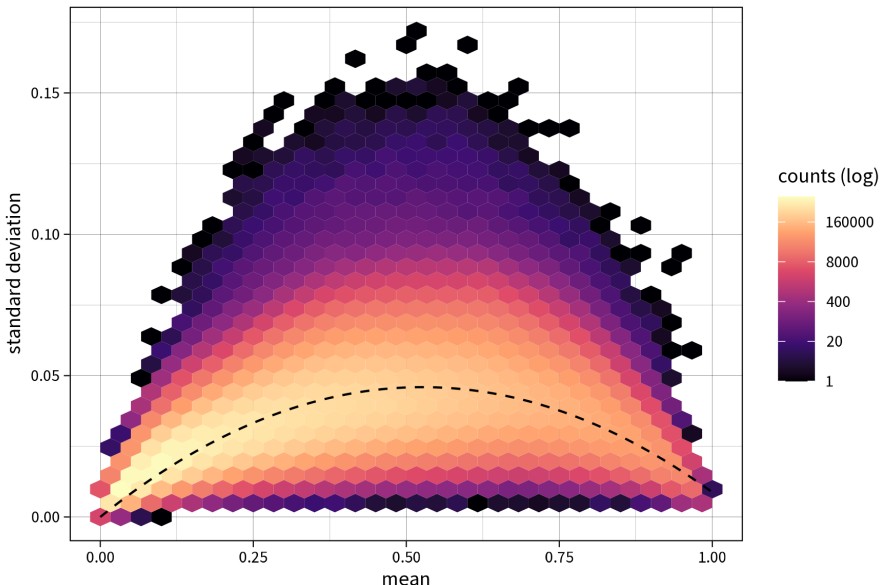

**Figure D1.** Ensemble mean versus ensemble standard deviation for all predictions of the full area of interest in Carinthia, Austria.

There are two main reasons for this emerging pattern. First, the standard deviation is bound to be lower towards both ends of the available susceptibility range for purely mathematical reasons. Since susceptibility values are confined between 0 and 1, the standard deviation is truncated at both ends of the spectrum. Naturally, this effect becomes more pronounced as the mean approaches either the minimum or maximum of the feature range. Second, predicted outcomes near the class discrimination threshold of 0.5 are typically more uncertain than instances classified as positive or negative with high confidence. In this medium range the model ensemble has difficulties classifying the terrain as either stable or unstable, which results in a higher standard deviation of the estimate and thus decreases the reliability of the prediction (Guzzetti et al., 2006).

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
