# Peer review of "Brief communication: Visualizing uncertainties in landslide susceptibility modeling using bivariate mapping"

_Natural Hazards and Earth System Sciences, 2024_

## Community Comment (CC1)

Review of:

"**Brief communication: Visualizing uncertainties in landslide susceptibility modeling using bivariate mapping**", Matthias Schlögl, Anita Graser , Raphael Spiekermann , Jasmin Lampert , and Stefan Steger. Natural Hazards and Earth System Sciences (NHESS),  https://doi.org/10.5194/nhess-2024-213

**1.  Related to susceptibility mapping of landslides**

This paper deals with a method for visualizing landslide susceptibility and associated uncertainty through bivariate mapping (randon forest modeling of landslide susceptibility and associated uncertainty, and bivariate mapping).

The model used (random forest) results in the mapping of susceptibility (means) and uncertainty (standard deviations), the combination of which (bivariate mapping) is the subject of this paper.

Landslide susceptibility modeling is based on the "randon forest" method (p. 3 of the article: 2.1 Landslide susceptibility modeling). The implementation of "randon forest" is preceded by the collection, selection and the classification of events (1973 events). What period/years do these events cover?

These events served as target variable (i.e., training labels): in the matrix used by the model (observations in rows and variables in columns), each event is represented by which parameters / characteristics: probability of occurrence, magnitude, damage, etc.?

It would be interesting to indicate the main determinants: elevation, slope, precipitation… (independent variables: Variation Inflation Factor VIF?) selected and the method used to select them (regression coefficient: Ordinary Least Square or other?) with regard to the dependent variable studied (landslide events).

"A more detailed description of the modeling approach as well as an in-depth discussion focusing on statistical performance and geomorphic plausibility is provided in Schlögl et al. (2024)[1]":

Finally, I propose to better describe both the study site (its dimensions: how many km in length and width, number of pixels in rows and columns, etc.) and the method for modeling landslide susceptibility, this method is apparently considered by another paper (Schlögl et al., 2024) but which is currently being evaluated. The aim is to show that the two maps (susceptibility and uncertainty) are based on a method that is both (statistically) validated and (methodologically) reproducible.
* * *
[1] Schlögl, M., Spiekermann, R., and Steger, S.: Towards a holistic assessment of landslide susceptibility models: Insights from the Central Eastern Alps, Environmental Earth Sciences, (Under review), 2024.

The landslide hazard is probably linked to other natural hazards (precipitation, freezing and thawing periods, etc.). Can't we talk about a landslide triggered by other hazards (multi-hazard)? Perhaps we're probably getting off the topic of this paper (Schlögl et al., 2024?).

Uncertainty can be taken into account at different stages/steps:

** when calculating susceptibility (if possible), resulting in only a single dataset (instead of two datasets: means and standard deviations) by "random forest" itself?

** in post-processing (if possible), by crossing the two datasets (means and standard deviations: like the coefficient of variation)?

** or by bivariate mapping, as proposed in the article (e.g. crossing the two rasters: Raster 1 of Susceptibility * Raster 2 of Uncertainty)

**2. Related to mapping of susceptibility and uncertainty**

"We advocate that bivariate mapping is a straightforward yet sound and effective way to communicate landslide susceptibility and the associated uncertainty."

How can we show/verify that the bivariate map is more effective in conveying the message than the two initial maps of susceptibility and uncertainty?

It's a question of visual perception and cognitive understanding of the final map by end-users (elected representatives, citizens, tourists), in order to confirm that the final map is more effective (or not).

When we consider 3 susceptibility classes and 3 uncertainty classes (3 x 3), we obtain 9 classes or 9 color gradations (bivariate map). If we go to 4 x 4, we'll have 16 color gradations, which makes reading the bivariate map even more & more complex…

Secondly, the use of the visual variable color (which can be aesthetic and attractive) certainly brings us closer to human visual perception, which is immediately colorful (and in 3D), but what mental realities do the color used represent of the landscape / site? Is it interesting to represent uncertainty in blue gradation color (high level of uncertainty in blue)?

The paper can try to present / propose a second color combinations.

As future development, the definitive choice of color used (gradation in one color for each of susceptibility and uncertainty) can be determined / confirmed ALSO with the help of end-users (students, researchers, laypersons, decision makers).

The 3D block diagrams in the appendices help to understand the results (visual link between the two variables: slope and high landslide susceptibility-uncertainty).

Is the scale of variation of susceptibility (between 0 and 1) different (or not) from that of uncertainty (standard deviations)? It is the determination of the limits considered for the creation of the 3 classes in both cases that raises the question here in terms of scales of variation of susceptibility and uncertainty. Has uncertainty been standardized?

Finally, here are references related to mapping aspect:

| [Bertin 1977] | Bertin J. (1977). La graphique et le traitement graphique de l'information, Flammarion, Paris, 277 p. |
|---|---|
| [Bertin 1983] | Bertin J., (1983) *Semiology of Graphics: Diagrams, Networks, Maps.* University of Wisconsin Press, (first published in French in 1967, translated to English by Berg W.J. in 1983) |
| [Board 1972)] | Board C. (1972). Cartographic communication. Cartographica: The International Journal for Geographic Information and Geovisualization, vol. 18, n°2, p. 42-78. |
| [Brewer 2003] | Brewer C. A, Hatchard G. W, Harrower M. A, (2003) ColorBrewer in print: a catalog of color schemes for maps. Cartography and Geographic Information Science. 30(1). 5–32. www.ColorBrewer.org |
| [Buard et al. 2007] | Buard E., Ruas A. Evaluation of colour contrasts by means of expert knowledge for on demand mapping, *Actes de la conférence de cartographie internationale ICA 2007*, Moscou. |
| [Fuchs 2009] | Fuchs S., (2009). Evaluating cartographic design in flood risk mapping. Environmental Hazards, vol. 8, p. 52-70. |
| [Hegarty et al. 2009] | Hegarty M. et al. (2009). Naïve cartography: how intuition about display configuration can hurt performance, Cartographica, vol. 44, n° 3,p. 171-186. |
| [Meyer et al. 2012] | Meyer V., Kuhlicke, C., Luther, J., Fuchs, S., Priest, S., Dorner, W., Serrhini, K., Pardoe, J., McCarthy, S., Seidel, J., Palka, G., Unnerstall, H., Viavattene, C., & Scheuer, S. (2012), Recommendations for the user-specific enhancement of flood maps, Natural Hazards Earth System. Science, vol. 12, n° 5, p. 1701–1716. |

Regards. K. Serrhini

---

## Author Response (AR1)

**nhess-2024-213 » Author's response**

> Schlögl, M., Graser, A., Spiekermann, R., Lampert, J., and Steger, S.: Brief communication: Visualizing uncertainties in landslide susceptibility modeling using bivariate mapping, Nat. Hazards Earth Syst. Sci. Discuss. [preprint], https://doi.org/10.5194/nhess-2024-213, in review, 2024.

**Response to editor comments**

Dear editor,

please find the reviewer comments and our responses below.

Based on your comment we have expanded several of our responses to address the reviewer comments more explicitly instead of primarily referring to another published paper or literature.

With respect to some comments raised in CC1 we would like to point out that we think that discussing technical details on the methodological modelling workflow of the dataset used for visualization purposes is beyond the scope of this manuscript. The main focus of this manuscript is the presentation of a visualization method. This presentation would work with different kinds of data sets, even with an artificially generated one. We figured that it would be nice to build upon a recently published result to showcase this method using an actual real-world example. Therefore, we provide answers to the questions raised by the reviewer below, but we did not include all these technical details in the revised version. We think that this would distract from the main focus of the manuscript, which is on uncertainty communication and visualizing uncertainties and not on discussing the intricacies of modelling apporach used for creating the underlying dataset. The latter topic, which is more technical in nature, is discussed in detail in the companion paper. We therefore argue that referring to the companion paper where the methodological details of the dataset used in this manuscript are described in detail does not compromise the relevance and scientific integrity of this manuscript.

We added the most important statements when referencing other work concisely in the revised version of this manuscript (e.g. by succinctly summarizing the term geomorphic plausibility, which is discussed extensively in other publications).

Best regards,
Matthias Schlögl

**Response to RC1**

> Using the Central Eastern Alps as an example, this brief communication presents a unique method that integrates the vulnerability and uncertainty of landslides into a single bivariate map. Compared to the conventional approach of examining landslide risk through separate maps, this method offers several advantages. First, by integrating two maps into one, readers can avoid the hassle of cross-referencing, which greatly improves efficiency and reduces the possibility of errors. Secondly, the bivariate map provides a more holistic and intuitive understanding of the complex interplay between vulnerability and uncertainty, enhancing the overall assessment of landslide risk.
>
> The manuscript is well-structured and professionally written. Therefore, the reviewer has no issues with agreeing to publication in its current form.

*Thank you for the positive feedback and for taking the time to review the manuscript. We are glad to hear that you found the manuscript well-structured and professionally written. We appreciate your recommendation for publication.*

**Response to RC2**

> The authors present a bivariate mapping method to spatially visualize both prediction values and uncertainty within the same map. They also provide supplemental material to enable others to apply this approach using free and open-source software packages. The authors discuss and present an approach for estimating uncertainties and classifying susceptibility levels required to build a bivariate map.
>
> This brief communication is well-written and provides a generally significant contribution to the landslide science community, as it highlights a method for improving the communication of uncertainties in hazards predictions. I would recommend minor revisions to address some literature gaps in the introduction and discussion.
>
> The paper fails to acknowledge previously peer-reviewed research on visualization methods for geospatial predictions. The only reference on this topic is from a blog post, which is problematic given that the main contribution of this paper is the visualization of spatial prediction uncertainties. In particular, it is missing references to highly cited research by MacEachren et al (2013 in Cartography and Geographic Information Science), earlier applications to slope stability (Davis and Keller 1997 in Computers & Geosciences), and others who have applied bivariate mapping for communicating spatial prediction and uncertainties (Cola 2013 in Cartography and Geographic Information Science and Nelson 1999 in Cartographic Perspectives).
>
> Another major drawback of this paper is its heavy reliance on reference to works by its own co-authors (Steger and Spiekermann) for landslide susceptibility modelling and data quality, while failing to acknowledge other important contributions in the field.

*Thanks for the constructive and helpful feedback provided in this reviewer comment.*

*We agree that there are some literature gaps in this article. To some extent, the brevity in some aspects is attributable to the manuscript type (i.e., "Brief Communication") and the corresponding guidelines in terms of manuscript length, number of figures and number of references (c.f. https://www.natural-hazards-and-earth-system-sciences.net/about/manuscript_types.html). We did already stretch the limits in terms of the number of references, but felt that a more extensive reference list was needed for this article. In our endeavor to keep the article length and reference list short, we have omitted some potentially interesting publications that could be cited here.*

*We agree with the reviewer that the papers mentioned are of high relevance and added references to the following publications as suggested by the reviewer:*

- *MacEachren et al (2005): https://doi.org/10.1559/1523040054738936*
- *Davis and Keller (1997): https://doi.org/10.1016/S0098-3004(97)00012-5*
- *DeCola (2013): https://doi.org/10.1559/152304002782008413*
- *Nelson (1999): https://doi.org/10.14714/CP32.625*

*Concerning the references by the co-authors, the article includes one reference to a paper by Spiekermann et al. and indeed four (well cited) references to manuscripts authored by Steger et al. Given the background of the senior author, whose research focus is on uncertainty assessment in landslide susceptiblity modelling since many years, we think that this number is not excessive.*

**Other comments**

**Introduction**

> The introduction provides a good overview of methods applied for spatially estimating uncertainties of landslide susceptibility predictions. However, it is lacking background on general methods for visualizing and communicating uncertainties in spatial predictions. Existing research on this topic should be incorporated to help position the authors' approach within the context of prior work.

*We added the following references related to visualizing and communicating uncertainties in spatial predictions to the introduction:*

- *Kinkeldey et al. (2014): http://doi.org/10.1179/1743277414Y.0000000099*
- *MacEachren et al. (2005): https://doi.org/10.1559/1523040054738936*
- *Davis and Keller (1997): https://doi.org/10.1016/S0098-3004(97)00012-5*

> L20. The authors rely heavy on citing the co-authors' prior contributions on data quality. However, there are many different researchers with significant contributions in this field, and these should be acknowledged.

*The focus on our prior contributions simply reflects our emphasis on data quality within previous research. We agree with your suggestion and have added references to other significant contributions to ensure broader acknowledgment:*

- *Gaidzik & Ramírez-Herrera (2021): https://doi.org/10.1038/s41598-021-98830-y*
- *Loche et al. (2022): https://doi.org/10.1016/j.earscirev.2022.104125*

> L40. The paper should also reference Heckmann et al. (2014 in NHESS), who used repeated resampling and combined (100) susceptibility maps to estimate the interquartile range (IQR) in spatially predicted probabilities – a similar approach to the one used in this submission.

*Thank you for this literature suggestion. We added a reference to Heckmann et al. (2014) https://doi.org/10.5194/nhess-14-259-2014. We noted that the metric used in this study is actually the interquantile range between the 0.95-quantile and the 0.05-quantile, thereby encompassing 90 % of the modelled susceptibility values.*

**Methods**

> Section 2.1: The authors cite a blog post as the source of their methods but fail to reference earlier peer-reviewed contributions using a similar approach (e.g., Cola et al. 2013). While the authors clarify that their uncertainty calculations are based solely on variations in the sampling of absence (non-landslide) points, given the large number of landslide samples (~2000), they could have also resampled landslides (e.g., using a cross-validation approach). At the very least, they should acknowledge that more robust approaches, which account for variations in landslide presence data, are available.

*Thanks for pointing this out. We fully agree and added the following references in this section:*

- *Trumbo (1981): http://doi.org/10.1080/00031305.1981.10479360*
- *Nelson (1999): https://doi.org/10.14714/CP32.625*

- *Speich et al. (2015): http://doi.org/10.1016/j.jhydrol.2015.01.086*
- *Teuling et al. (2011): http://doi.org/10.1002/joc.2153*
- *DeCola (2013): https://doi.org/10.1559/152304002782008413*

*Since the main purpose of this brief communication is on the visualization aspect, we did not discuss the underlying models in detail. We agree that acknowledging that more robust approaches exist is important and added a corresponding remark in the text.*

> Section 2.2.1, L87. I think it's good that you acknowledge the source of your inspiration for your bivariate approach; however, existing peer-review on bivariate mapping approaches to communicate prediction and uncertainty should also be acknowledged.

*Thanks for pointing this out. We fully agree and updated the references in this section.*

**Results**

> L133. The authors mention the term "geomorphic plausibility" in the introduction, but don't define it. I think it would be useful to the reader to define it.

*Thank you for this suggestion. In our previous paper, we introduced the term geomorphic plausibility in the context of landslide susceptibility modelling (cf. Steger et al. 2016). In summary, a geomorphic plausibility assessment serves as an evidence-based approach to identify implausible predictions, akin to the concept of 'biological plausibility' used to evaluate empirical relationships in biology. It aims to detect whether a landslide susceptibility map reflects evident errors or artefacts from the classification algorithm or input data, contradicting geomorphic plausibility. Following also Oreskes et al. (1994), a model is geomorphically implausible if the resulting map exhibits detectable flaws. This subjective evaluation is supported by holistic interpretations of exploratory data analyses, modelled relationships, and the spatial structure of predictors and predicted patterns. In order to maintain the focus of this "brief communication type publication," we refer the reader to the mentioned publication and expanded this section as follows: "Geomorphic plausibility evaluation aims to assess whether a landslide susceptibility map aligns with fundamental process knowledge or rather reflects errors stemming from input data or the modeling approach, as detailed in Steger et al. (2016)"*

**Discussion**

> L159. "In addition, the type of uncertainty conveyed should be kept in mind" – what are you trying to communicate with this sentence? It's not clear.

*There are different sources of uncertainty, notably including uncertainties stemming from the landslide inventory, explanatory features or the modelling algorithm used. We expanded this paragraph by appending the following sentence to make this more explicit: "In addition, the type of uncertainty conveyed should be kept in mind: The underlying uncertainties can be aleatoric and epistemic in nature. Especially epistemic uncertainty, stemming from different sources along the modelling workflow, including the inventory, explanatory features or the model used, can be visualized."*

> L164. "In landslide susceptibility modeling, results are commonly discretized into three classes signifying low, medium and high susceptibility" – this statement is highly debatable. There are a wide range of approaches in practice to classify landslide susceptibility levels.

*We agree and removeed the word 'three', indicating more generally that a discretization of the continuous variable into susceptibility classes is commonly performed.*

> L166. The authors again cite only co-authors' works (Spiekermann and Steger), even though there are other well-cited approaches for calculating breaks (e.g. Chung and Fabbri 2003) including accounting for proportion of landslides covered (e.g. Petschko et al 2014), as applied in this submission.

*With respect to the derivation of class intervals we did cite the following publications (L162f): Slocum et al. (2022), Jiang (2013), Jenks and Caspall (1971) as well as Fisher (1958). We supplemented this section with the following additional references:*

- *Chung and Fabbri (2003): http://doi.org/10.1023/B:NHAZ.0000007172.62651.2b*
- *Costanzo et al. (2012): http://doi.org/10.5194/nhess-12-327-2012*
- *Conoscenti et al. (2016): http://doi.org/10.1016/j.geomorph.2016.03.006*
- *Petschko et al. (2014): http://doi.org/10.5194/nhess-14-95-2014*
- *Hussin et al. (2016): http://doi.org/10.1016/j.geomorph.2015.10.030*

> L175. The authors discuss the importance of color considerations but should expand on how their chosen color palette addresses issues like color impairment (e.g., for colorblind readers). Providing recommendations for alternative palettes or best practices would be helpful for readers.

*In terms of recommendations for alternative palettes we refer to the R package biscale, which implements a set of bivariate mapping palettes. The palettes provided therein are based on recommended palettes used for map representations. We added the twelve color palettes provided in biscale as well as map representations with six selected palettes in the appendix. In addition, tools such as https://colorbrewer2.org or https://hclwizard.org (c.f. the R package `colorspace`) provide valuable assistance for selecting palettes for users with different types of vision deficiencies. We added a reference to these two tools as a footnote.*

> L185. The authors should reference other approaches for quantifying uncertainties in landslide susceptibility models to provide a more balanced discussion.

*We supplemented this statement with the following references to provide a broader context:*

- *Rossi et al. (2010): http://doi.org/10.1016/j.geomorph.2009.06.020*
- *Petschko et al. (2014): http://doi.org/10.5194/nhess-14-95-2014*
- *Brenning et al. (2015): https://doi.org/10.5194/nhess-15-45-2015*
- *Lombardo et al. (2020): https://doi.org/10.1016/j.earscirev.2020.103318*

**Response to CC1**

*Thank you for the input and the suggestions. Given that we the manuscript under consideration is a "Brief Communication" we feel that several of the remarks exceed the scope of this type of manuscript. We therefore refer to Schlögl et al. (2025), where the questions raised here are answered, and the study area, data, methods and methodology is described and discussed in detail. The companion paper is available at [https://doi.org/10.1007/s12665-024-12041-y](https://doi.org/10.1007/s12665-024-12041-y). In this manuscript, we decided to summarize the core aspects relevant to obtain a brief overview and general understanding of the data visualized in the bivariate map. While the underlying model is of course important, we wanted to focus on the method of presenting and communicating estimates and associated uncertainty via bivariate mapping, using a recent model as a case study example. This is also beneficial for providing data alongside the code in the supplementary GitHub repository. Please find more detailed responses below.*

**1. Related to susceptibility mapping of landslides**

> This paper deals with a method for visualizing landslide susceptibility and associated uncertainty through bivariate mapping (randon forest modeling of landslide susceptibility and associated uncertainty, and bivariate mapping). The model used (random forest) results in the mapping of susceptibility (means) and uncertainty (standard deviations), the combination of which (bivariate mapping) is the subject of this paper.
>
> Landslide susceptibility modeling is based on the "randon forest" method (p. 3 of the article: 2.1 Landslide susceptibility modeling). The implementation of "randon forest" is preceded by the collection, selection and the classification of events (1973 events). What period/years do these events cover?

*The inventory used includes all landslides documented by October 2023. Please note that not all recorded events have an exact timestamp (i.e., information on the exact event day is not always available, especially for events farther back in time). Many events in the inventory were mapped using a very high resolution digital terrain model, which is continuously updated (every five years), whith different areas of Carinthia being covered in regular intervals. The earliest events date back to the 1920s and 1930s, the first quartile is 1996, the median 2009, and the third quartile 2007. The newest events are from 2023. The empirical density of all events with date information looks as follows:*

**Empirical density of landslide records with date info**

[Figure]

> These events served as target variable (i.e., training labels): in the matrix used by the model (observations in rows and variables in columns), each event is represented by which parameters / characteristics: probability of occurrence, magnitude, damage, etc.?

*Since the model is a binary classification model, each event is simply characterized by its presence. Event size is available for many events (i.e., events are mapped as polygons), but due to the regional scale of the analysis events are summarized to data points signifying the landslide scarp. Damage, magnitude or recurrence interval are not considered for the susceptibility model. Information on physical damage is not available.*

> It would be interesting to indicate the main determinants: elevation, slope, precipitation... (independent variables: Variation Inflation Factor VIF?) selected and the method used to select them (regression coefficient: Ordinary Least Square or other?) with regard to the dependent variable studied (landslide events).

Detailed information on the main determinants are available from the feature importance plot of the underlying model as provided in Schlögl et al. (2025). The main determinants include the vector ruggedness measure, land cover type, slope, tree height and the 30-day SPEI (a drought indicator). Please find correlation heatmaps for (1) all features initially considered and (2) the reduced feature set used for modelling. For the second (reduced) feature set, features that exhibit redundant information as indicated by high Pearson correlation coefficients ($\rho > 0.6$) were omitted.

[Figure]

[Figure]

> "A more detailed description of the modeling approach as well as an in-depth discussion focusing on statistical performance and geomorphic plausibility is provided in Schlögl et al. (2024)" Finally, I propose to better describe both the study site (its dimensions: how many km in length and width, number of pixels in rows and columns, etc.) and the method for modeling landslide susceptibility, this method is apparently considered by another paper (Schlögl et al., 2024) but which is currently being evaluated. The aim is to show that the two maps (susceptibility and uncertainty) are based on a method that is both (statistically) validated and (methodologically) reproducible.

*The size of the study area is stated right at the beginning of subsection 2.1:*

"The study region encompasses an area of $5,785$ km² within the Central Eastern Alps in Carinthia, Austria."

*The size of the sub-area used for visualization is 999 times 499 pixels. The pixel size is 10x10 meters (in EPSG:3416). The full data set and the corresponding outline of the area of interest are provided in the supplementary GitHub repository at https://github.com/r3xth0r/bivariate-lsm.*

*We argue that the statistical method employed for assessing landslide susceptibility is of secondary importance in this context. The primary focus of this article is on the communication of uncertainties through bivariate mapping. The methodological approach and the corresponding discussion of the model fall outside the scope of this article.*

*The maps are fully reproducible, since all data used for creating the maps as well as the corresponding R scripts and QGIS-Project are available in the supplementary GitHub repository. In addition, all resources related to the landslide susceptibility model (including all scripts for data preprocessing, data preparation, sampling, model tuning, training and validation) are available on GitLab at https://gitlab.com/Rexthor/lsm-carinthia. These resources are mentioned in the "Code and data availability statement" at the end of the manuscript.*

> The landslide hazard is probably linked to other natural hazards (precipitation, freezing and thawing periods, etc.). Can't we talk about a landslide triggered by other hazards (multi-hazard)? Perhaps we're probably getting off the topic of this paper (Schlögl et al., 2024?).

*This is of course a relevant topic. However, the focus of this manuscript lies on visualization of results in terms of science-to-science as well as science-to-public communication. Additional discussions related to landslide mechanics, geomorphology and multizahard impact assessment are beyond the scope of this manuscript.*

> Uncertainty can be taken into account at different stages/steps:
>
> - when calculating susceptibility (if possible), resulting in only a single dataset (instead of two datasets: means and standard deviations) by "random forest" itself?
> - in post-processing (if possible), by crossing the two datasets (means and standard deviations: like the coefficient of variation)?
> - or by bivariate mapping, as proposed in the article (e.g. crossing the two rasters: Raster 1 of Susceptibility * Raster 2 of Uncertainty)

*This is correct. We state in the article that "[t]he uncertainty displayed here merely refers to the estimation uncertainty stemming from the sampling of negative instances, as quantified through the ensemble modeling approach." Please also see our response to RC2 with respect to epistemic and aleatory uncertainties, and the corresponding changes to the manuscript.*

**2. Related to mapping of susceptibility and uncertainty**

> "We advocate that bivariate mapping is a straightforward yet sound and effective way to communicate landslide susceptibility and the associated uncertainty." How can we show/verify that the bivariate map is more effective in conveying the message than the two initial maps of susceptibility and uncertainty? It's a question of visual perception and cognitive understanding of the final map by end-users (elected representatives, citizens, tourists), in order to confirm that the final map is more effective (or not).

*We did conduct stakeholder workshops (world cafe method) with selected expert users from the main target group (including geologists from the Austrian national geological survey, geologists from Austrian federal governments and representatives from disaster relief forces), who provided unstructured feedback for using the map a specific context and for specific tasks. While the users needed some time to become accustomed to the visualization initially, once they internalized the legend the combined map was considered to be more effective. The main reasons were that (1) no switching between map views was required, which makes the joint interpretation less cumbersome (i.e., lower cognitive effort for internalizing the legend once is lower than switching between maps), and (2) the consideration of the second dimension (uncertainty) was considered less likely to be neglected. However, a formalized usability evaluation (such as qualitative usability tests with users or usability expert reviews conducted by usability professionals, or quantitative assessments such as A/B tests) or studies conducted by cognitive psychologists would be required to formally verify the effectiveness of this method. Unfortunately, this is beyond the scope of the study. We expanded the methods and results section accordingly.*

> When we consider 3 susceptibility classes and 3 uncertainty classes (3 x 3), we obtain 9 classes or 9 color gradations (bivariate map). If we go to 4 x 4, we'll have 16 color gradations, which makes reading the bivariate map even more & more complex...

*This is true. Arguably, the interpretability when using 4 x 4 classes is lower despite the additional amount of information contained in the map, as the 16 different gradiations are more difficult to distinguish visually. Therefore, our results are based on 9 classes, which can be distinguished comparatively easily when using appropriate color palettes while still providing enough information on the core messages conveyed by the map.*

> Secondly, the use of the visual variable color (which can be aesthetic and attractive) certainly brings us closer to human visual perception, which is immediately colorful (and in 3D), but what mental realities do the color used represent of the landscape / site? Is it interesting to represent uncertainty in blue gradation color (high level of uncertainty in blue)? The paper can try to present / propose a second color combinations. As future development, the definitive choice of color used (gradation in one color for each of susceptibility and uncertainty) can be determined / confirmed ALSO with the help of end-users (students, researchers, laypersons, decision makers).

*The choice of color palette is briefly discussed in the discussion section (line 174ff). We emphasize the importance of accessibility (especially for people with vision deficiencies), the cultural and contextural relevance, and point towards employing a user-centered design process for tailoring visualized information to users. Modifying the color palette is straightforward from a technical point of view, as only the string specifying the name of the palette has to be changed in R. In QGIS, the color palette can easily be changed via the GUI in the properties of the raster layer. In addition to predefined color palettes, custom palettes could be provided relatively easily. We have created several alternative representations for demonstration purposes. The provided these in the appendix.*

> The 3D block diagrams in the appendices help to understand the results (visual link between the two variables: slope and high landslide susceptibility-uncertainty).

*We also think that rayshader animations provide additional insights by adding information on the terrain. We added an interactive rayshader visualization as supplementary material, and serve the animation via GitHub pages at [https://r3xth0r.github.io/bivariate-lsm/rayshader.html](https://r3xth0r.github.io/bivariate-lsm/rayshader.html).*

> Is the scale of variation of susceptibility (between 0 and 1) different (or not) from that of uncertainty (standard deviations)? It is the determination of the limits considered for the creation of the 3 classes in both cases that raises the question here in terms of scales of variation of susceptibility and uncertainty. Has uncertainty been standardized?

*Both scales refer to the same dimensionless quantity, namely landslide susceptibility, proxied by the classification probability of the underlying binary classification problem. Susceptibility is the mean of the ensemble, uncertainty its corresponding standard deviation. We do discuss the issue of deriving univariate class intervals in continuous numerical vectors in the discussion section (line 160 ff).*

---

## Author Response (AR2)

**Editor Comments**

> As you have included information on stakeholder workshops in the revised version, it is essential to provide additional details, such as institutional ethics approval, consent procedures, and risk assessment conducted before the workshops. Additionally, please specify when, where and how the workshops took place and the number of participants involved. Kindly add a few sentences in the Methods section where this information has been introduced.
>
> The rest of the manuscript looks good to me.

**Response**

We have updated the Methods section to include details on informed consent, the location and time of the workshops, as well as the number of participants involved. Additionally, we mentioned that the workshops were organized and moderated by representatives of the Disaster Competence Network Austria (DCNA), whose strategic goal is to bridge the gap between scientific research and practitioners in crisis and disaster management in Austria. We have also acknowledged the contribution of DCNA in a new Acknowledgements section at the end of the manuscript. Furthermore, letters of support from the majority of institutions that participated in the workshops were collected during the funding acquisition process. Workshop participants attended as experts representing their respective institutions and public authorities, rather than as private individuals. Due diligence was exercised throughout the process to ensure the accuracy and integrity of the information provided.